# Generalized Category Discovery via Reciprocal Learning and Class-Wise Distribution Regularization

**Duo Liu** [1]  **Zhiquan Tan** [2]  **Linglan Zhao** [# 1]  **Zhongqiang Zhang** [1]  **Xiangzhong Fang** [1]  **Weiran Huang** [† 1 3 4]

## Abstract

Generalized Category Discovery (GCD) aims to identify unlabeled samples by leveraging the base knowledge from labeled ones, where the unlabeled set consists of both base and novel classes. Since clustering methods are time-consuming at inference, parametric-based approaches have become more popular. However, recent parametric-based methods suffer from inferior base discrimination due to unreliable self-supervision. To address this issue, we propose a Reciprocal Learning Framework (RLF) that introduces an auxiliary branch devoted to base classification. During training, the main branch filters the pseudo-base samples to the auxiliary branch. In response, the auxiliary branch provides more reliable soft labels for the main branch, leading to a virtuous cycle. Furthermore, we introduce Class-wise Distribution Regularization (CDR) to mitigate the learning bias towards base classes. CDR essentially increases the prediction confidence of the unlabeled data and boosts the novel class performance. Combined with both components, our proposed method, RLCD, achieves superior performance in all classes with negligible extra computation. Comprehensive experiments across seven GCD datasets validate its superiority. Our codes are available at https://github.com/APORduo/RLCD.

## 1. Introduction

With the development of deep learning in recent years, models can perform well in traditional tasks such as image recognition (He et al., 2016; 2017; Vaswani et al., 2017;

[1]School of Electronic Information and Electrical Engineering, Shanghai Jiao Tong University [2]Department of Mathematical Sciences, Tsinghua University [3]Shanghai Innovation Institute [4]State Key Laboratory of General Artificial Intelligence, BIGAI. [#]Project lead. [†]Correspondence to: Weiran Huang <weiran.huang@outlook.com>.

*Proceedings of the 42nd International Conference on Machine Learning*, Vancouver, Canada. PMLR 267, 2025. Copyright 2025 by the author(s).

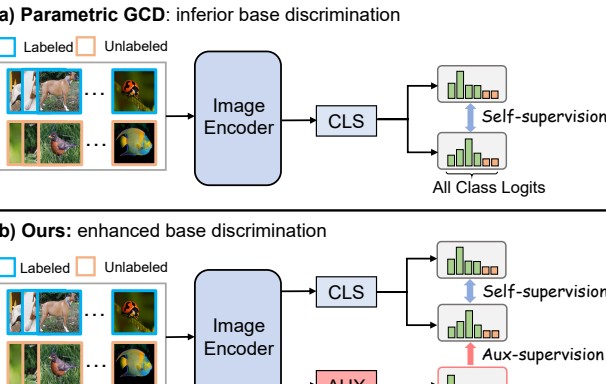

**(a) Parametric GCD**: inferior base discrimination

**(b) Ours**: enhanced base discrimination

*Figure 1.* (a) Parametric Generalized Category Discovery (GCD) methods rely on self-supervision for clustering unlabeled data but exhibit weak base class discrimination. (b) Our approach introduces an auxiliary branch specialized in base class classification, providing more reliable base logits to the main branch and thereby significantly improving base discrimination.

Dosovitskiy, 2021). Generally, the models rely on abundant annotated data in a closed scenario where the unlabeled data share the same classes with the labeled training data. However, these models have limitations in the real-world scenario where unlabeled data comes from unknown classes. In this way, Category Discovery (CD) has garnered attention in the machine learning community. Initially, Han et al. (2019) propose Novel Class Discovery (NCD), which is designed to cluster novel class data with the assistance of labeled data exclusively. However, NCD assumes that the unlabeled data all belong to novel classes, which is unrealistic in practical scenarios. Recently, Generalized Category Discovery (GCD) (Vaze et al., 2022b) has emerged and it allows the unlabeled data spanning both base and novel categories. Compared to the NCD task, GCD is more practical and challenging in real-world scenarios.

Vaze et al. (2022b) first define the GCD problem and tackle it using contrastive learning along with the semi-supervised $k$-means clustering method. Wen et al. (2023)further propose an effective parametric framework, SimGCD, which outperforms the clustering methods with reduced inference time. Due to its effectiveness, the parametric framework

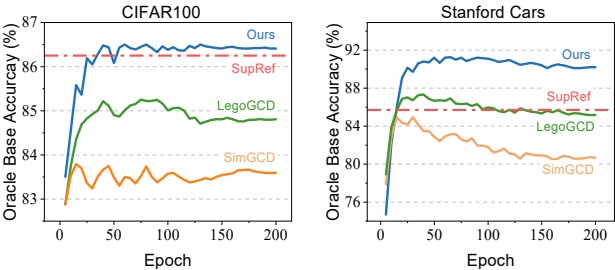

*Figure 2.* Comparison of the oracle base class accuracy between SimGCD, LegoGCD, and our method. SimGCD and LegoGCD exhibit poor performance, falling behind the supervised reference (SupRef). Contrarily, our method exhibits enhanced discrimination, even surpassing SupRef.

has become popular in GCD research. Wang et al. (2024) design a two-stage framework on the pre-trained SimGCD model that introduces both global and spatial prompts to fine-tune the model. LegoGCD(Cao et al., 2024) observes that SimGCD suffers catastrophic forgetting of base classes, and they propose a novel regularization to address it. Despite the significant advancements in parametric methods, Fig. 1 reveals a key limitation: these methods rely exclusively on self-supervision, leading to suboptimal base discrimination.

To quantitatively reveal the main limitation in existing works, we define the *oracle base accuracy* (OB) for evaluating base discrimination. OB solely considers base-class prediction and calculates the accuracy of unlabeled base data utilizing the prototype classifier. As shown in Fig. 2, both SimGCD and LegoGCD lag behind the supervised-only reference (SupRef), which exclusively utilizes labeled data for training. This disparity primarily arises from the unreliable soft labels in self-supervised learning.

To promote base discrimination, we design a Reciprocal Learning Framework (RLF). In particular, we insert an auxiliary token named AUX in the model architecture. Then AUX is concatenated with the CLS token and image feature tokens to form the input of the final block. Subsequently, the corresponding AUX output is dedicated to a base-only classifier while the CLS output is designated for the all-class classifier. During training, the main branch filters pseudo-base samples, which are predicted to the base classes, and directs them to the auxiliary branch. In feedback, the auxiliary branch provides reliable base class distribution to the main branch. This collaboration between the two branches contributes to more robust base predictions, improving base-class discrimination and overall accuracy. Benefiting from parallel computation between tokens, the extra computation cost is negligible.

However, the reciprocal framework may incur learning bias toward base classes that more novel samples are misclassified into the base classes. To alleviate the above bias, we propose a Class-wise Distribution Regularization (CDR) technique. Specifically, CDR involves calculating the expected distribution for each category based on mini-batch predictions. Then, CDR loss promotes expectation consistency between two views of the mini-batch and boosts prediction confidence. Since each class can be treated equally, CDR effectively mitigates the bias and boosts novel class performance. By integrating CDR into the RLF, our method, named RLCD, obtains substantial improvements.

Our key contributions can be summarized as follows:

- We define the oracle base class accuracy (OB) as a metric to assess the base class discrimination of GCD models, revealing the inferior discrimination of parametric GCD methods.

- We design a novel reciprocal learning framework to promote base class discrimination and a class-wise distribution regularization loss to improve novel class performance.

- Experimental results on seven GCD datasets show that the proposed method consistently outperforms state-of-the-art approaches in most scenarios.

## 2. Related Works

**Semi-Supervised Learning (SSL)** is a prominent area in machine learning that addresses the challenge of training models with limited labeled data. Pseudo Label (Lee et al., 2013) iteratively assigns pseudo labels for unlabeled data, which join the labeled set for further training. Mean-teacher (Tarvainen & Valpola, 2017), UDA (Xie et al., 2020), Fixmatch (Sohn et al., 2020) adopt confidence threshold to generate pseudo labels on weak augmented samples and utilize it to supervise strongly augmented samples, and DST (Chen et al., 2022) proposes an adversary framework to refine pseudo labels. ConMatch (Kim et al., 2022) adds self-supervised features regularization, while SimMatch (Zheng et al., 2022) extends consistency to the semantic and instance levels. Prevailing semi-supervised methods widely adopt threshold-based pseudo-label learning during training. However, this mechanism faces significant limitations when unlabeled data includes samples from unknown classes.

**Novel Class Discovery (NCD)** aims to recognize novel classes in unlabeled data by exploiting knowledge from known classes. (Han et al., 2019) first proposes the NCD problem and addresses it utilizing a two-stage training strategy. (Han et al., 2020) employ rank statistics to find positive data pairs and pull them closer. OpenMix (Zhu et al., 2023) generates virtual samples by MixUp between labeled and unlabeled data, guiding the model to resist noisy labeled data. (Zhong et al., 2021) proposes neighborhood contrastive learning to aggregate pseudo-positive pairs. (Fini et al., 2021) introduces a unified objective framework with

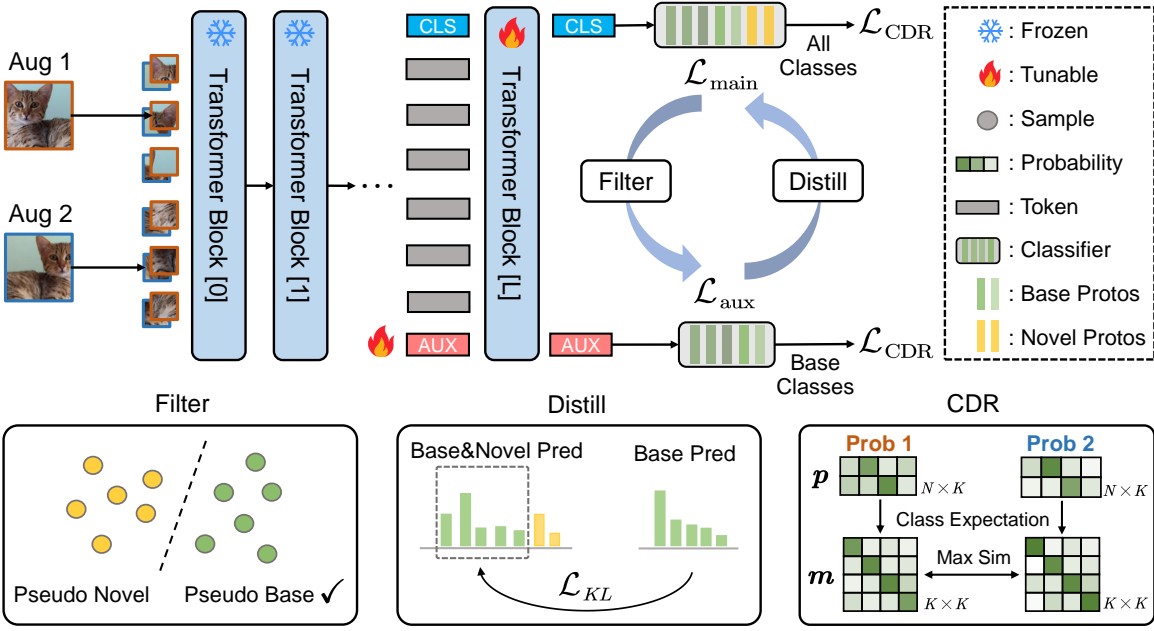

*Figure 3.* Overview of our method. We insert an auxiliary token `AUX` before the last block of the ViT backbone. The final `AUX` feature is utilized for the base-only classifier while the `CLS` feature is assigned to the all-class classifier. The main branch filters the pseudo-base samples to the aux branch for better base class learning. In response, the auxiliary branch provides the main branch with the refined base class distribution. Class-wise Distribution Regularization (CDR) boosts novel performance by maximizing the similarity between class-wise probability matrices $\boldsymbol{m}$ from two views.

the Sinkhorn-Knopp algorithm, allowing cross-entropy to operate on both labeled and unlabeled sets. CRNCD (Gu et al., 2023) conducts a class-relationship distillation approach to improve novel-class performance. However, this distillation shows inferior performance on GCD. Unlike CRNCD, we propose a novel one-stage distillation method tailored for GCD.

**Generalized Category Discovery (GCD)** is to cluster unlabeled images by leveraging the base knowledge from labeled images, where the unlabeled set comprises both base and novel classes. (Vaze et al., 2022b) formulates the GCD problem and conducts contrastive training on a pre-trained ViT model (Dosovitskiy, 2021) with DINO (Caron et al., 2021), followed by semi-supervised $k$-means clustering. Recent works have extended GCD to settings such as active learning (Ma et al., 2024b) and continual learning (Ma et al., 2024a). CiPR (Hao et al., 2024) designs a novel contrastive learning method by exploiting cross-instance positive relations in labeled data and introducing a hierarchical clustering algorithm. GPC (Zhao et al., 2023a) applies Gaussian mixture models that learn robust representation and estimate the novel class number. Wen et al. (2023) propose a parametric framework that trains a prototype classifier to fit all categories. SimGCD utilizes mean-entropy regularization to automatically find novel classes. As SimGCD boots GCD performance with lower inference latency, the parametric framework becomes popular. SPTNet (Wang et al., 2024)

introduces a two-stage strategy that combines the global and spatial prompts to further finetune the SimGCD model. Lin et al. (2024) design a teacher-student attention alignment strategy to promote GCD performance. LegoGCD (Cao et al., 2024) finds that SimGCD suffers from catastrophic forgetting in training and solves it by adding regularization to potential known class samples. While parametric-based methods achieve great GCD performance, they often experience degraded base discrimination. To address this issue, we propose a Reciprocal Learning Framework (RLF) that provides more reliable base pseudo-labels and effectively strengthens base performance.

## 3. Method

### 3.1. Preliminaries

**Problem formulation.** Generalized Category Discovery (GCD) aims to adaptively cluster unlabeled data utilizing the knowledge from labeled data. GCD is built upon the open-world dataset, which compromises two subsets: labeled dataset $\mathcal{D}^l = \{(\boldsymbol{x}_i, y_i)\} \in \mathcal{X} \times \mathcal{Y}^l$ and unlabeled dataset $\mathcal{D}^u = \{(\boldsymbol{x}_i, y_i)\} \in \mathcal{X} \times \mathcal{Y}^u$. Formally, $\mathcal{Y}^l$ is a subset of $\mathcal{Y}^u$, and $\mathcal{Y}^u$ spans all categories. Following previous research, the number of $|\mathcal{Y}^u|$ is assumed as the prior. GCD adopts a transductive training strategy in which all the samples are involved in the training process.

**Parametric clustering.** (Wen et al., 2023) proposes an effi-

cient parametric framework that builds a prototype classifier for clustering. Specifically, the classifier weight is the set of prototypes $\mathcal{C} = \{c_1, \ldots, c_K\}$, where $K$ is the total number of prototypes. Given an image $x_i$, the model correspondingly output feature $f(x_i)$, and the probability of category $k$ is denoted as:

$$p_i^{(k)} = \frac{\exp\left(\cos\left(f\left(x_i\right), c_k\right) / \tau_s\right)}{\sum_{k'} \exp\left(\cos\left(f\left(x_i\right), c_{k'}\right) / \tau_s\right)}, \quad (1)$$

where $\cos$ denotes the cosine similarity between two vectors and $\tau_s$ is the temperature scalar. Similarly, the shrink probability $q_i$ can be derived by substituting $\tau_s$ with a smaller $\tau_t$. Subsequently, SimGCD adopts the cross entropy loss $\mathcal{L}_{\text{ce}}(q, p) = -\sum_k q^{(k)} \log p^{(k)}$ to regularize the probability self-consistency between two views of an image. For $i$-th image, the loss is formulated as:

$$\mathcal{L}_{\text{self}}^{(i)} = \frac{1}{2}\mathcal{L}_{\text{ce}}\left(q'_i, p_i\right) + \frac{1}{2}\mathcal{L}_{\text{ce}}\left(q_i, p'_i\right), \quad (2)$$

where $p'_i$ and $q'_i$ are the prototype probabilities of another view. Additionally, SimGCD employs a mean-entropy maximization regulariser for clustering: $H(\overline{p}) = -\sum_k \overline{p}^{(k)} \log \overline{p}^{(k)}$ where $\overline{p}$ is the mean predicted probability of all the samples. The supervised loss for the labeled data is the sum of cross-entropy and supervised contrastive learning losses (Khosla et al., 2020):

$$\mathcal{L}_{\text{con}}^{(i)} = \frac{1}{|\mathcal{P}_i|} \sum_{q \in \mathcal{P}_i} -\log \frac{\exp\left(\cos(f\left(x_i\right), f(x'_q)) / \tau_c\right)}{\sum_{n \neq i} \exp\left((f\left(x_i\right), f\left(x'_n\right)) / \tau_c\right)},$$
$$\mathcal{L}_{\text{sup}}^{(i)} = \mathcal{L}_{\text{ce}}\left(y_i, p_i\right) + \mathcal{L}_{\text{con}}^{(i)}, \quad (3)$$

where $y_i$ is the one-hot distribution associated with $y_i$, $\tau_c$ denotes the temperature scalar for supervised contrastive learning and the $\mathcal{P}_i$ is the positive index set sharing the same label as $x_i$. While SimGCD applies InfoNCE (Oord et al., 2018) loss in the training, we found the loss tends to push apart same-class features, which conflicts with the SupCon loss and impairs feature discrimination. Consequently, we chose to remove InfoNCE in our approach for better discrimination. Overall, the parametric clustering loss $\mathcal{L}_{\text{cls}}$ is the average per-sample combination of supervised loss, self-consistency loss, and entropy regularization loss:

$$\mathcal{L}_{\text{cls}} = \lambda\mathcal{L}_{\text{sup}} + (1 - \lambda)(\mathcal{L}_{\text{self}} - \epsilon H(\overline{p})), \quad (4)$$

where $\lambda$ is the balance weight belonging to [0,1] and $\epsilon$ is the scalar to control entropy regularization.

### 3.2. Reciprocal Learning Framework

**Motivation.** While SimGCD demonstrates greater effectiveness than clustering methods, it falls short in base class discrimination. Specifically, when focusing

on base classification, the unlabeled base data is defined as $\mathcal{D}_{\text{base}}^u = \{(x_i, y_i) | (x_i, y_i) \in \mathcal{D}^u, y_i \in \mathcal{Y}^l\}$. The oracle base accuracy is defined as $ACC_{\text{OB}} = \frac{1}{|\mathcal{D}_{\text{base}}^u|} \sum_{x_i, y_i \in \mathcal{D}_{\text{base}}^u} \mathbb{1}\left(\widetilde{y}_i = y_i\right)$, where $\widetilde{y}_i$ is the predicted base class result. As depicted in Fig. 2, prevailing parametric methods exhibit unsatisfactory oracle base accuracy, falling behind the supervised-only reference.

To this end, we propose a one-stage Reciprocal Learning Framework (RLF). As shown in Fig. 3, we insert the auxiliary token AUX before the last block, concatenating it with CLS and feature tokens to form the input. The final AUX feature is utilized for the base-only classification, while the CLS feature is assigned to the all-class classifier. Different from the CLS feature, which is unique to each image, the AUX token is a trainable parameter shared across all training samples.

During the training procedure, the main branch is akin to generic parametric clustering. Besides, the main branch filters the pseudo-base class samples to the auxiliary branch according to the prediction result, *i.e.*, if a sample is predicted to belong to the base classes, it will also be involved in the auxiliary branch. In response, the auxiliary branch distills the base class prediction of pseudo-base samples to the main branch. The collaboration between the two branches effectively enhances base discrimination, mitigates the influence of noise labels, and facilitates the model in acquiring improved representations.

Note that most of the training samples will be predicted as the base classes initially, and the auxiliary branch also incorporates abundant novel samples at the same time. To this end, the auxiliary branch adopts self-supervised learning and supervised learning, rather than threshold-based semi-supervised methods. Furthermore, we utilize the maximum probability as the uncertainty weight for each pseudo-base sample in the cross-branch distillation. The distillation loss for a pseudo-base sample $i$ is denoted as:

$$\mathcal{L}_{\text{dis}}^{(i)} = \max(p_{b,i}^{\text{aux}}) \cdot \mathcal{L}_{\text{KL}}(p_{b,i}^{\text{aux}}, p_{b,i}), \quad (5)$$

where $p_b^{\text{aux}}$, $p_b$ is the base class distribution from the auxiliary and main branch, $\mathcal{L}_{\text{KL}}$ is the standard KL-divergence loss, and the auxiliary probability is detached in the distillation. Consequently, the loss functions of the two branches can be presented as:

$$\mathcal{L}_{\text{main}} = \mathcal{L}_{\text{cls}} + \alpha\mathcal{L}_{\text{dis}}, \ \mathcal{L}_{\text{aux}} = \mathcal{L}_{\text{sup}} + \mathcal{L}_{\text{self}}, \quad (6)$$

where $\alpha$ controls the distillation strength.

### 3.3. Class-Wise Distribution Regularization.

While the proposed reciprocal framework can effectively improve base class discrimination, it still shows inferior performance in the novel classes. This is primarily due to

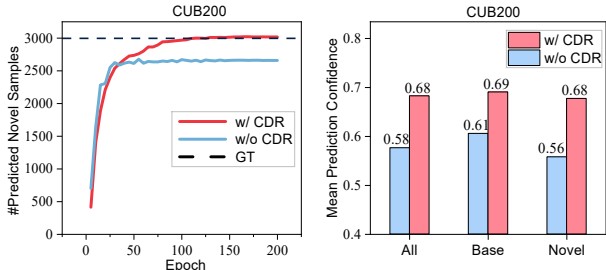

*Figure 4.* The efficacy of CDR is evident in two aspects. Left: CDR induces more predicted novel class samples. Right: CDR contributes to higher prediction confidence.

the cross-branch distillation being confined to base class distributions, resulting in a learning bias where training samples are more likely to be recognized as base classes. Fig. 4 (a) shows that the predicted novel samples lag behind the ground truth number. To mitigate the learning bias, we propose a novel Class-wise Distribution Regularization (CDR) shown in the bottom right of Fig. 3.

Adopted by (Zhang et al., 2024), the class-wise expected distribution $m$ for category $k$ is defined as:

$$\boldsymbol{m}_k = \frac{1}{\sum_{i=1}^{N} \boldsymbol{p}_i^{(k)}} \left( \sum_{i=1}^{N} \boldsymbol{p}_i^{(k)} \boldsymbol{p}_i \right), \qquad (7)$$

where $N$ is the batch size, and $m$ has a dimension of $K \times K$. For the main branch, $K^{\text{main}}$ is the number of all classes, while $K^{\text{aux}}$ denotes the number of base classes in the auxiliary branch. Intuitively, the $k$-th probability of $m_k$, denoted as $m(k, k)$, reflects the confidence that "the mini-batch contains at least one sample belonging to category $k$."

**Theorem 3.1.** *The sum of all elements in $\boldsymbol{m}_k$ equals 1, i.e.,* $\mathbf{1}^T \boldsymbol{m}_k = 1$ *(Zhang et al., 2024).*
Theorem 3.1 shows that $m_k$ conforms to the standard probability distribution. Intuitively, the class-wise prediction should be consistent between the two views of the images and close to the one-hot distribution. To this end, for class $k$, the CDR loss is formulated as

$$\mathcal{L}_{\text{CDR}}^{(k)} = 1 - \langle \boldsymbol{m}_k, \boldsymbol{m}_k' \rangle, \qquad (8)$$

where $\langle \cdot, \cdot \rangle$ denotes the inner product calculation, representing the similarity between two distributions, and $\boldsymbol{m}_k'$ is the expectation from another view. Since each class is treated equally, CDR effectively alleviates the bias towards the base classes in the main branch.

**Theorem 3.2.** $\mathcal{L}_{\text{CDR}}^{(k)}$ *equals to zero* $\iff$ $\boldsymbol{m}_k$ *equals* $\boldsymbol{m}_k'$ *and is a one-hot distribution.*

*Proof.* Please refer to the Appendix A.

Theorem 3.2 indicates that CDR essentially increases the prediction confidence, approaching the one-hot distribution.

The effectiveness of CDR is evidenced in Fig. 4, as it leads to a higher number of predicted novel class samples, reducing learning bias and boosting prediction confidence. Furthermore, the CDR loss operates independently of ground-truth labels and is compatible with both branches. When applied to the auxiliary branch, CDR also benefits base class learning with minimal impact on the novel class performance of the main branch. By integrating the CDR loss into the reciprocal framework, the overall loss is summarized as:

$$\mathcal{L} = \mathcal{L}_{\text{main}} + \mathcal{L}_{\text{aux}} + \beta \mathcal{L}_{\text{CDR}}, \qquad (9)$$

where $\beta$ controls the regularization weight. After the training procedure, we abandon the auxiliary classifier and only keep the main branch for evaluation. As a result, the inference latency difference from SimGCD is negligible.

## 4. Experiments

### 4.1. Experimental Setup

**Datasets.** Following previous works, we evaluate our method on seven different GCD datasets. Those consist of generic image recognition datasets CIFAR10/100 (Krizhevsky et al., 2009) and ImageNet-100 (Tian et al., 2020); Semantic Shit Benchmark (SSB) (Vaze et al., 2022a) datasets: CUB200 (Wah et al., 2011), Stanford Cars (Krause et al., 2013), and FGVC-Aircraft (Maji et al., 2013); large-scale fine-grained dataset: Herbarium-19 (Tan et al., 2019). Formally, each dataset is partitioned into base and novel subsets. The novel subset data is entirely unlabeled, while half of the base data is labeled during training, with the remaining half left unlabeled. For a fair comparison, we adopt the same random seed in the data split with (Vaze et al., 2022b).

**Evaluation metric.** We adopt cluster accuracy ($ACC$) to evaluate the performance of our method. More specifically, given the samples' prediction $\hat{y}$ and ground-truth labels $y$, the Hungarian optimal assignment algorithm (Kuhn, 1955) allocates the clustering result and calculates the accuracy. $ACC = \frac{1}{|\mathcal{D}^u|} \sum_{i=1}^{|\mathcal{D}^u|} \mathbb{1}\left(y_i = \mathcal{G}\left(\hat{y}_i\right)\right)$, where $\mathcal{G}$ denotes the optimal permutation function.

**Implementation details.** In alignment with prevailing GCD methods, we conduct our experiments using the pre-trained DINO (Caron et al., 2021) ViT-B/16 and DINOv2 (Oquab et al., 2024) ViT-B/14 backbones. Unless otherwise specified, we adopt ViT-B/16 for experimental analysis. Training parameters involve the last block and the auxiliary token across all datasets. The final output retains the features from the CLS and AUX tokens for classification. The default learning rate is set to 0.1, following a cosine annealing decay schedule. Our model is trained for 200 epochs with a batch size of 128. Following (Wen et al., 2023), the temperature scalars are $\tau_c = 0.1, \tau_s = 0.07$, while $\tau_t$ scales from 0.07

*Table 1.* Comparative results on the Semantic Shift Benchmark and Herbarium-19.

| Method | Backbone | CUB200 | | | Stanford Cars | | | FGVC-Aircraft | | | Herbarium-19 | | | Avg. |
|---|---|---|---|---|---|---|---|---|---|---|---|---|---|---|
| | | All | Base | Novel | All | Base | Novel | All | Base | Novel | All | Base | Novel | |
| $k$-means (Macqueen, 1967) | DINO | 34.3 | 38.9 | 32.1 | 12.8 | 10.6 | 13.8 | 16.0 | 14.4 | 16.8 | 13.0 | 12.2 | 13.4 | 19.0 |
| RS+ (Han et al., 2021) | DINO | 33.3 | 51.6 | 24.2 | 28.3 | 61.8 | 12.1 | 26.9 | 36.4 | 22.2 | 27.9 | 55.8 | 12.8 | 29.1 |
| UNO+ (Fini et al., 2021) | DINO | 35.1 | 49.0 | 28.1 | 35.5 | 70.5 | 18.6 | 40.3 | 56.4 | 32.2 | 28.3 | 53.7 | 14.7 | 34.8 |
| ORCA (Cao et al., 2022) | DINO | 35.3 | 45.6 | 30.2 | 23.5 | 50.1 | 10.7 | 22.0 | 31.8 | 17.1 | 20.9 | 30.9 | 15.5 | 25.4 |
| ∗CRNCD (Gu et al., 2023) | DINO | 62.7 | 71.6 | 58.2 | 54.1 | 75.7 | 43.7 | 54.4 | 59.5 | 51.8 | 41.3 | 60.7 | 30.9 | 53.1 |
| GCD (Vaze et al., 2022b) | DINO | 51.3 | 56.6 | 48.7 | 39.0 | 57.6 | 29.9 | 45.0 | 41.1 | 46.9 | 35.4 | 51.0 | 27.0 | 42.7 |
| DCCL (Pu et al., 2023) | DINO | 63.5 | 60.8 | 64.9 | 43.1 | 55.7 | 36.2 | - | - | - | - | - | - | - |
| GPC (Zhao et al., 2023a) | DINO | 55.4 | 58.2 | 53.1 | 42.8 | 59.2 | 32.8 | 46.3 | 42.5 | 47.9 | 36.5 | 51.7 | 27.9 | 45.3 |
| SimGCD (Wen et al., 2023) | DINO | 60.3 | 65.6 | 57.7 | 53.8 | 71.9 | 45.0 | 54.2 | 59.1 | 51.8 | 44.0 | 58.0 | 36.4 | 53.1 |
| uGCD (Vaze et al., 2024) | DINO | 65.7 | 68.0 | 64.6 | 56.5 | 68.1 | 50.9 | 53.8 | 55.4 | 53.0 | - | - | - | - |
| CMS (Choi et al., 2024) | DINO | 68.2 | 76.5 | 64.0 | 56.9 | 76.1 | 47.6 | 56.0 | **63.4** | 52.3 | 36.4 | 54.9 | 26.4 | 54.4 |
| InfoSieve (Rastegar et al., 2024) | DINO | 69.4 | 77.9 | 65.2 | 55.7 | 74.8 | 46.4 | 56.3 | 63.7 | 52.5 | 41.0 | 55.4 | 33.2 | 55.6 |
| SPTNet (Wang et al., 2024) | DINO | 65.8 | 68.8 | 65.1 | 59.0 | 79.2 | 49.3 | 59.3 | 61.8 | 58.1 | 43.4 | 58.7 | 35.2 | 56.9 |
| LegoGCD (Cao et al., 2024) | DINO | 63.8 | 71.9 | 59.8 | 57.3 | 75.7 | 48.4 | 55.0 | 61.5 | 51.7 | 45.1 | 57.4 | 38.4 | 55.3 |
| **RLCD (Ours)** | DINO | **70.0** | **79.1** | **65.4** | **64.9** | **79.3** | **58.0** | **60.6** | 62.2 | **59.8** | **46.4** | **61.2** | **38.4** | **60.5** |
| SimGCD (Wen et al., 2023) | DINOv2 | 74.9 | 78.5 | 73.1 | 71.3 | 81.6 | 66.4 | 63.9 | 69.9 | 60.9 | 58.7 | 63.8 | 56.2 | 67.2 |
| uGCD (Vaze et al., 2024) | DINOv2 | 74.0 | 75.9 | 73.1 | 76.1 | 91.0 | 68.9 | 66.3 | 68.7 | 65.1 | - | - | - | - |
| CiPR (Hao et al., 2024) | DINOv2 | 78.3 | 73.4 | **80.8** | 66.7 | 77.0 | 61.8 | - | - | - | 59.2 | 65.0 | **56.3** | - |
| **RLCD (Ours)** | DINOv2 | **78.7** | **79.5** | 78.3 | **79.5** | **91.8** | **73.5** | **72.6** | **77.3** | **70.3** | **60.2** | **71.9** | 54.0 | **72.8** |

to 0.04 within 30 epochs, and the balance weight $\lambda = 0.35$. The default hyper-parameters in our method are specified as $\alpha = 0.5, \beta = 0.5$. The augmentation includes Resize, RandomCrop, Random Horizontal Flip, Color Jittering, and Image Normalization. All experiments are conducted on a single NVIDIA GeForce 3090 GPU based on PyTorch.

## 4.2. Main Results

We compare our approach with SOTA methods including clustering-based methods: $k$-means (Macqueen, 1967), GCD (Vaze et al., 2022b), GPC (Zhao et al., 2023a), DCCL (Pu et al., 2023), uGCD (Vaze et al., 2024) InfoSieve (Rastegar et al., 2024), CMS (Choi et al., 2024); parametric-based methods: SimGCD (Wen et al., 2023), SPTNet (Wang et al., 2024), LegoGCD (Cao et al., 2024) and strong baseline derived from NCD: RS+ (Han et al., 2021), UNO+ (Fini et al., 2021), ORCA (Cao et al., 2022), CRNCD (Gu et al., 2023). The best results are highlighted in bold and ∗ denotes reproduced results.

**Evaluation on fine-grained datasets.** Table 1 shows the comparative results on four fine-grained datasets which are more challenging than the generic. Clustering methods demonstrate inadequate performance, falling far behind the parametric methods on average. The proposed RLCD consistently outperforms the others across all four datasets and two backbones, with a notable 3.6% average improvement on DINO. Its strong base performance is attributed to reliable base logits from the auxiliary branch, while its superior novel class accuracy results benefit from the effective regularization loss.

**Evaluation on generic datasets.** As shown in Table 2, we present the comparison on generic datasets including CIFAR10/100 and ImageNet-100. While RLCD performs on par with existing methods on CIFAR10, it achieves the best results on CIFAR100 and ImageNet-100. As a result, RLCD attains the highest average accuracy, improving by 0.8% and 1.6% on DINO and DINOv2, respectively. Notably, our method surpasses the two-stage SPTNet while utilizing fewer parameters during training. Note that the backbone is pre-trained on the extensive generic dataset like ImageNet-1k (Deng et al., 2009). Leveraging the strong feature representation of pretrained backbones, existing methods perform comparably on datasets such as CIFAR10 and ImageNet-100. Additionally, the abundance of labeled data in these three datasets minimizes discrimination degradation in parametric methods.

**Comparison of OB performance.** As defined in Section 3.2, OB serves as a metric to evaluate the base class discrimination capability of GCD models. As depicted in Fig. 5, the proposed RLCD method achieves the best OB performance across all seven GCD datasets, with particularly significant improvements on fine-grained datasets. The comparison further highlights the great discrimination of our method.

## 4.3. Ablation Study

**Effect of different loss components.** As previously outlined, our approach mainly has four loss components: main branch loss (Main), auxiliary branch loss (AUX), cross-branch distillation (Distill), and class-wise distribution regularization (CDR). Here we demonstrate their effectiveness on CIFAR100 and CUB200 datasets. Additionally, we introduce the oracle base class accuracy (OB) as an additional

*Table 2.* Comparative results on generic image recognition datasets.

| Methods | Backbone | CIFAR10 | | | CIFAR100 | | | ImageNet-100 | | | Avg. |
|---|---|---|---|---|---|---|---|---|---|---|---|
| | | All | Base | Novel | All | Base | Novel | All | Base | Novel | |
| $k$-means (Macqueen, 1967) | DINO | 83.6 | 85.7 | 82.5 | 52.0 | 52.2 | 50.8 | 72.7 | 75.5 | 71.3 | 69.4 |
| RS+ (Han et al., 2021) | DINO | 46.8 | 19.2 | 60.5 | 58.2 | 77.6 | 19.3 | 37.1 | 61.6 | 24.8 | 47.4 |
| UNO+ (Fini et al., 2021) | DINO | 68.6 | **98.3** | 53.8 | 69.5 | 80.6 | 47.2 | 70.3 | 95.0 | 57.9 | 69.5 |
| ORCA (Cao et al., 2022) | DINO | 81.8 | 86.2 | 79.6 | 69.0 | 77.4 | 52.0 | 73.5 | 92.6 | 63.9 | 74.8 |
| ∗CRNCD (Gu et al., 2023) | DINO | 96.9 | 97.5 | 96.6 | 80.3 | 84.7 | 71.5 | 81.4 | 94.4 | 74.8 | 86.2 |
| GCD (Vaze et al., 2022b) | DINO | 91.5 | 97.9 | 88.2 | 73.0 | 76.2 | 66.5 | 74.1 | 89.8 | 66.3 | 79.5 |
| DCCL (Pu et al., 2023) | DINO | 96.3 | 96.5 | 96.9 | 75.3 | 76.8 | 70.2 | 80.5 | 90.5 | 76.2 | 84.0 |
| GPC (Zhao et al., 2023a) | DINO | 92.2 | 98.2 | 89.1 | 77.9 | 85.0 | 63.0 | 76.9 | 94.3 | 71.0 | 82.3 |
| SimGCD (Wen et al., 2023) | DINO | 97.1 | 95.1 | 98.1 | 80.1 | 81.2 | 77.8 | 83.0 | 93.1 | 77.9 | 86.7 |
| InfoSieve (Rastegar et al., 2024) | DINO | 94.8 | 97.7 | 93.4 | 78.3 | 82.2 | 70.5 | 80.5 | 93.8 | 73.8 | 84.5 |
| CiPR (Hao et al., 2024) | DINO | **97.7** | 97.5 | 97.7 | 81.5 | 82.4 | 79.7 | 80.5 | 84.9 | 78.3 | 86.6 |
| CMS (Choi et al., 2024) | DINO | - | - | - | 82.3 | **85.7** | 75.5 | 84.7 | **95.6** | 79.2 | - |
| SPTNet (Wang et al., 2024) | DINO | 97.3 | 95.0 | **98.6** | 81.3 | 84.3 | 75.6 | 85.4 | 93.2 | 81.4 | 88.0 |
| LegoGCD (Cao et al., 2024) | DINO | 97.1 | 94.3 | 98.5 | 81.8 | 81.4 | **82.5** | 86.3 | 94.5 | 82.1 | 88.4 |
| **RLCD (Ours)** | DINO | 97.4 | 96.4 | 97.9 | **83.4** | 84.2 | 81.9 | **86.9** | 94.2 | **83.2** | **89.2** |
| SimGCD (Wen et al., 2023) | DINOv2 | 98.8 | 96.9 | **99.7** | 88.5 | 89.3 | 86.9 | 88.5 | 96.2 | 84.6 | 91.9 |
| CiPR (Hao et al., 2024) | DINOv2 | 99.0 | 98.7 | 99.2 | 90.3 | 89.0 | **93.1** | 88.2 | 87.6 | 88.5 | 92.5 |
| **RLCD (Ours)** | DINOv2 | **99.0** | **98.9** | 99.1 | **91.2** | **91.2** | 91.2 | **92.1** | **96.2** | **90.0** | **94.1** |

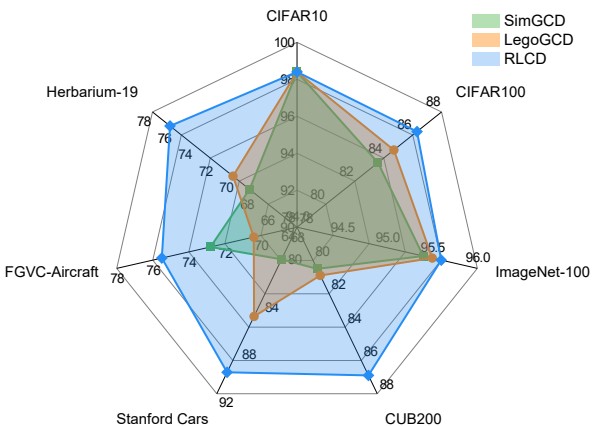

*Figure 5.* Comparison of oracle base accuracy among SimGCD, LegoGCD, and our RLCD.

metric for evaluation.

Table 3 shows the GCD performance with different loss configurations. Specifically, using only the Main loss in (a) serves as the baseline, while (f) represents our complete method. The comparison of (a) with (b) indicates that CDR enhances overall performance with minimal influence on OB, contributing a substantial 5.3% novel class improvement on CUB200. The addition of the AUX loss in (c) leads to comprehensive performance gains and improved discrimination. Since the final block is shared between the two branches, the model learns more robust parameters to fit different tasks, resulting in enhanced feature representation. Comparing (c) with (d) validates that distillation further strengthens performance, improving base accuracy by 2.2%.

*Table 3.* Ablation experiments on different configurations of loss components: Main, AUX, Distill, and CDR. OB denotes the Oracle base class accuracy and $M$ represents only on main branch.

| ID | Main | AUX | Distill | CDR | CUB200 | | | |
|---|---|---|---|---|---|---|---|---|
| | | | | | All | Base | Novel | OB |
| (a) | ✓ | | | | 62.1 | 70.8 | 57.7 | 83.9 |
| (b) | ✓ | | | $M$ | 65.7 | 71.1 | 63.0 | 83.7 |
| (c) | ✓ | ✓ | | | 64.3 | 73.7 | 59.7 | 85.9 |
| (d) | ✓ | ✓ | ✓ | | 66.6 | 76.4 | 61.7 | 86.4 |
| (e) | ✓ | ✓ | ✓ | $M$ | 69.5 | 78.1 | 65.2 | 86.7 |
| (f) | ✓ | ✓ | ✓ | ✓ | **70.0** | **79.1** | 65.4 | **86.9** |
| (g) | ✓ | ✓ | | ✓ | 67.6 | 71.7 | **65.6** | 85.8 |
| (h) | ✓ | | ✓ | $M$ | 65.9 | 73.8 | 61.9 | 84.6 |

Building on (d), applying CDR to the main branch in (e) improves novel class performance, while applying it to both branches in (f) slightly enhances base class discrimination, yielding the best overall results. It turns out that CDR helps improve base class performance without exacerbating the base class learning bias. When either the Distill (g) or the AUX (h) is exclusively removed, our model suffers a significant drop in base performance. This outcome highlights the necessity of both components in our approach.

**Hyper-parameter sensitivity analysis.** As indicated in Equations (6) and (9), we utilize $\alpha$ and $\beta$ to control the distillation and regularization strength. Fig. 6 (a) illustrates the GCD performance curves with varying values of $\alpha$. As $\alpha$ increases, we observe a significant improvement in base class accuracy, aligning with the intuition that stronger distillation enhances base class dissemination. However, when $\alpha$ becomes excessively large, it leads to degradation in novel

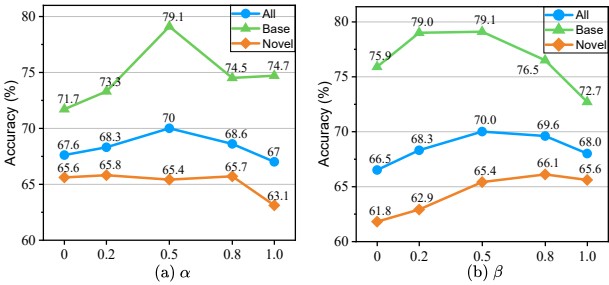

*Figure 6.* Effect of different weights of $\alpha$ and $\beta$ on CUB200.

class performance, ultimately harming overall accuracy. As shown in Fig. 6 (b), we see that increasing the weight of $\beta$ notably impacts novel class performance. However, overly large $\beta$ shows a negative effect on base class accuracy. Our analysis indicates that the optimal values for $\alpha$ and $\beta$ are approximately 0.5, which yields the best overall performance.

**Effect of different regularization.** We here present the GCD performance across different probability regularized methods. The baseline is our reciprocal learning framework (None) along with the comparative methods, including entropy minimization(ENT) (Grandvalet & Bengio, 2004), minimum class confusion (MCC) (Jin et al., 2020), and label-encoding risk minimization (LERM) (Zhang et al., 2024). Besides, we modify the CDR loss into pair-wise distribution regularization (PDR) that directly maximizes the probability similarity between two views of a sample.

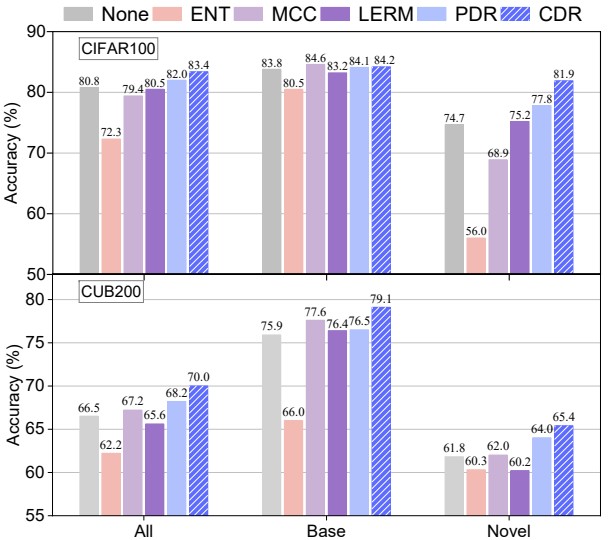

*Figure 7.* Comparison of various regularization methods on CI-FAR100 and CUB200, where CDR achieves the best results.

Fig. 7 indicates that prevailing regularization methods are not competitive in the GCD task. ENT shows a serious negative effect, leading to considerable degradation in both base and novel classes. MCC effectively improves the base performance yet harms the novel performance in both datasets.

LERM shows marginal effects as its performance remains close to the baseline. While PDR provides benefits on both datasets, the improvements are limited and exacerbate the performance gap between base and novel classes. In contrast, our proposed CDR effectively boosts novel class accuracy while maintaining base performance. Overall, the comparison demonstrates our proposed CDR is more appropriate for GCD tasks.

### 4.4. Further Analysis

**AUX branch is a good teacher.** We conduct a comprehensive analysis of base-class pseudo-label accuracy across all datasets, comparing our auxiliary branch (AUX), main branch (CLS), and the SimGCD baseline. The dataset names are the abbreviations according to all seven GCD datasets.

*Table 4.* Comparison of base-class pseudo-label accuracy.

|  | C10 | C100 | IN100 | CUB | Scars | Aircraft | Her19 |
|---|---|---|---|---|---|---|---|
| SimGCD | 98.4 | 83.6 | 95.4 | 80.5 | 80.7 | 72.8 | 68.6 |
| RLCD (CLS) | 98.4 | 86.3 | 95.6 | 86.9 | 90.2 | 75.5 | 76.3 |
| RLCD (AUX) | **98.6** | **87.1** | **96.2** | **88.3** | **91.4** | **75.9** | **76.4** |

From Table 4, the auxiliary branch shows the best pseudo-label accuracy across all datasets. The superior performance of AUX can be attributed to its specialized focus on base class discrimination, which leads to more reliable pseudo-labels. Consequently, this enhanced reliability directly contributes to the main branch's ability to maintain strong base class performance during training.

**Performance under estimated category number.** As the previous evaluation is built on the known category numbers $K$, we here report the results with estimated categories borrowed from off-the-shelf methods GCD (Vaze et al., 2022b) and GPC (Zhao et al., 2023a). As shown in Table 5, our method demonstrates slightly reduced performance on ImageNet-100, CUB200, and Stanford Cars when using GCD's estimation, yet still outperforms other methods. This performance drop can be attributed to the overestimation of $K$, which leads to unlabeled samples being clustered into a larger number of clusters. Notably, the impact on CUB200 is minimal, with only a 1.6% degradation. By leveraging the advanced estimation algorithm from GPC, which exhibits more accurate estimation, the performance gap is significantly reduced across all datasets. The differences are only 0.6%, 0.4%, and 2.5% compared to the ground-truth reference. The result indicates that our approach is not reliant on exact category numbers.

**Effect of varying ratios of labeled samples.** In the default experiment, 50% samples are labeled following the mainstream GCD setting. Here we explore the effect of various ratios of labeled samples on model performance.

Table 5. Comparison of estimated category numbers on ImageNet-100, CUB200, and Stanford Cars.

| Methods | $K$ | ImageNet-100 | | | CUB200 | | | Stanford Cars | | |
|---|---|---|---|---|---|---|---|---|---|---|
| | | All | Base | Novel | All | Base | Novel | All | Base | Novel |
| **RLCD (Ours)** | 100 / 200 / 196 | 86.9 | 94.2 | 83.2 | 70.0 | 79.1 | 65.4 | 64.9 | 79.3 | 58.0 |
| GCD (Vaze et al., 2022b) | 109 / 231 / 230 | 73.8 | 92.1 | 64.6 | 49.2 | 56.2 | 46.3 | 36.3 | 56.6 | 25.9 |
| SimGCD (Wen et al., 2023) | | 81.1 | 90.9 | 76.1 | 61.0 | 66.0 | 58.6 | 49.1 | 65.1 | 41.3 |
| SPTNet (Wang et al., 2024) | | 83.4 | 91.8 | 74.6 | 65.2 | 71.0 | 62.3 | - | - | - |
| **RLCD (Ours)** | | **84.4** | **93.2** | **80.0** | **68.4** | **77.1** | **64.1** | **58.6** | **76.4** | **50.8** |
| GPC (Zhao et al., 2023a) | 103 / 212 / 201 | 75.3 | 93.4 | 66.7 | 52.0 | 55.5 | 47.5 | 38.2 | 58.9 | 27.4 |
| **RLCD (Ours)** | | **86.3** | **94.1** | **82.4** | **69.6** | **78.3** | **65.2** | **62.4** | **78.1** | **54.8** |

Table 6. Performance comparison of RLCD and SimGCD under varying labeled sample proportions on CUB200.

| | Label Ratio (%) | All | Base | Novel |
|---|---|---|---|---|
| SimGCD | 50 | 60.3 | 65.6 | 57.7 |
| **RLCD (Ours)** | | **70.0** | **79.1** | **65.4** |
| SimGCD | 25 | 51.0 | 52.8 | 49.7 |
| **RLCD (Ours)** | | **63.8** | **67.5** | **61.1** |
| SimGCD | 10 | 34.6 | 31.8 | 37.3 |
| **RLCD (Ours)** | | **46.7** | **45.3** | **48.0** |

Table 6 reveals a clear performance degradation pattern as the labeled data ratio decreases. However, our RLCD maintains a significant performance advantage even with severely limited labeled data. The performance gap between RLCD and SimGCD widens in the low-resource scenario, demonstrating our method's superior capability in leveraging limited supervision.

**Effect of varying novel class numbers.** By default, 50% categories are selected as the novel class, following the standard GCD experiment setting. Here, we investigate the impact of varying the proportion of novel classes on model effectiveness, using SimGCD as the baseline for comparison.

Table 7. Performance comparison of RLCD and SimGCD with different novel class proportions on CUB200.

| | Novel Ratio(%) | All | Base | Novel |
|---|---|---|---|---|
| SimGCD | 50 | 60.3 | 65.6 | 57.7 |
| **RLCD (Ours)** | | **70.0** | **79.1** | **65.4** |
| SimGCD | 60 | 56.2 | 65.1 | 53.3 |
| **RLCD (Ours)** | | **62.5** | **75.6** | **58.1** |
| SimGCD | 75 | 51.8 | 68.3 | 49.1 |
| **RLCD (Ours)** | | **56.8** | **78.1** | **53.2** |

Seen from Table 7, our method consistently outperforms SimGCD across all metrics as the proportion of novel classes increases. The narrowing gap in novel class results

is expected, given the increased difficulty of clustering with more novel categories. Meanwhile, the strong results on base classes are largely maintained due to the reduced number of base categories. These results highlight the robustness of our approach under varying class distributions.

## 5. Conclusion

In this paper, we propose a novel approach for promoting generalized category discovery performance. To enhance base class discrimination in parametric clustering, we introduce the Reciprocal Learning Framework (RLF), which consists of two collaborative branches: an auxiliary branch that provides reliable soft labels, and a main branch that filters pseudo-base samples using an all-class classifier. Additionally, to mitigate the learning bias towards base classes, we further present Class-wise Distribution Regularization (CDR), which significantly boosts the prediction confidence of unlabeled data and strengthens the discovery of novel classes. The two components are complementary and together constitute our RLCD method. Extensive experiments confirm the effectiveness of RLCD, achieving great performance on both base and novel classes.

## Acknowledgment

This project is supported by the National Natural Science Foundation of China (No. 62406192), Opening Project of the State Key Laboratory of General Artificial Intelligence (No. SKLAGI2024OP12), Tencent WeChat Rhino-Bird Focused Research Program, and Doubao LLM Fund. The authors also thank Beijing PARATERA Tech CO., Ltd. (paratera.com) for HPC support.

## Impact Statement

This paper presents work whose goal is to advance the field of Generalized Category Discovery. There are many potential societal consequences of our work, none of which we feel must be specifically highlighted here.

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

# Appendix

## A. Theoretical Support

**Proof of Theorem 3.1**

*Proof.* $\mathbf{1}^T \boldsymbol{m}_k = \frac{\mathbf{1}^T}{\sum_{i=1}^{N} \boldsymbol{p}_i^{(k)}} \left( \sum_{i=1}^{N} \boldsymbol{p}_i^{(k)} \boldsymbol{p}_i \right) = \frac{\left( \sum_{i=1}^{N} \boldsymbol{p}_i^{(k)} (\mathbf{1}^T \boldsymbol{p}_i) \right)}{\sum_{i=1}^{N} \boldsymbol{p}_i^{(k)}} = \frac{\sum_{i=1}^{N} \boldsymbol{p}_i^{(k)}}{\sum_{i=1}^{N} \boldsymbol{p}_i^{(k)}} = 1.$

**Proof of Theorem 3.2**

*Proof.* Let $\mathbf{a}$ and $\mathbf{b}$ be two probability distribution vectors in $\mathbb{R}^n$ to present $\boldsymbol{m}_k$ and $\boldsymbol{m}_k'$ :

$$\mathbf{a} = [a_1, a_2, \ldots, a_n], \quad \mathbf{b} = [b_1, b_2, \ldots, b_n]$$

subject to the constraints:

$$\sum_{i=1}^{n} a_i = 1, \quad \sum_{i=1}^{n} b_i = 1, \quad a_i \geq 0, \quad b_i \geq 0 \text{ for } i = 1, 2, \ldots, n.$$

The inner product is given by:

$$\mathbf{a} \cdot \mathbf{b} = \sum_{i=1}^{n} a_i b_i.$$

By the Cauchy-Schwarz inequality, we have:

$$\left( \sum_{i=1}^{n} a_i b_i \right)^2 \leq \left( \sum_{i=1}^{n} a_i^2 \right) \left( \sum_{i=1}^{n} b_i^2 \right).$$

Since $\sum_{i=1}^{n} a_i = 1$ and $\sum_{i=1}^{n} b_i = 1$, we can observe:

$$\sum_{i=1}^{n} a_i^2 \leq \sum_{i=1}^{n} a_i = 1, \sum_{i=1}^{n} b_i^2 \leq \sum_{i=1}^{n} b_i = 1.$$

Thus, we have:

$$\left( \sum_{i=1}^{n} a_i b_i \right)^2 \leq 1 \cdot 1 = 1 \implies \sum_{i=1}^{n} a_i b_i \leq 1.$$

For equality $\sum_{i=1}^{n} a_i b_i = 1$ to hold, the Cauchy-Schwarz inequality must achieve equality, which occurs if and only if $a_i$ and $b_i$ are linearly dependent:

$$a_i = c b_i \text{ for some constant } c \text{ for all } i.$$

Given the constraints $\sum_{i=1}^{n} a_i = 1$ and $\sum_{i=1}^{n} b_i = 1$, it follows that:

$$1 = c \sum_{i=1}^{n} b_i = c \cdot 1 \implies c = 1.$$

Therefore, we have:

$$a_i = b_i \text{ for all } i.$$

Besides,

$$\sum_{i=1}^{n} a_i^2 \leq \sum_{i=1}^{n} a_i = 1 \implies a_i \in \{0, 1\}$$

which means:

$$a_j = 1 \text{ for some } j \text{ and } a_i = 0 \text{ for } i \neq j.$$

Thus, we conclude that:

$$\mathbf{a} = \mathbf{b} \text{ and both are one-hot distributions.}$$

## B. Dataset Split

As shown in Fig. 8, we illustrate the dataset split of Generalized Category Discovery(GCD) and compare it with Semi-Supervised Learning (SSL) and Novel Class Discovery (NCD). SSL assumes the labeled and unlabeled data share the same classes, NCD suggests unlabeled data all form the novel classes, while GCD allows the unlabeled data to belong to all classes. The comparison indicates GCD task is more challenging and practical in real-world scenarios.

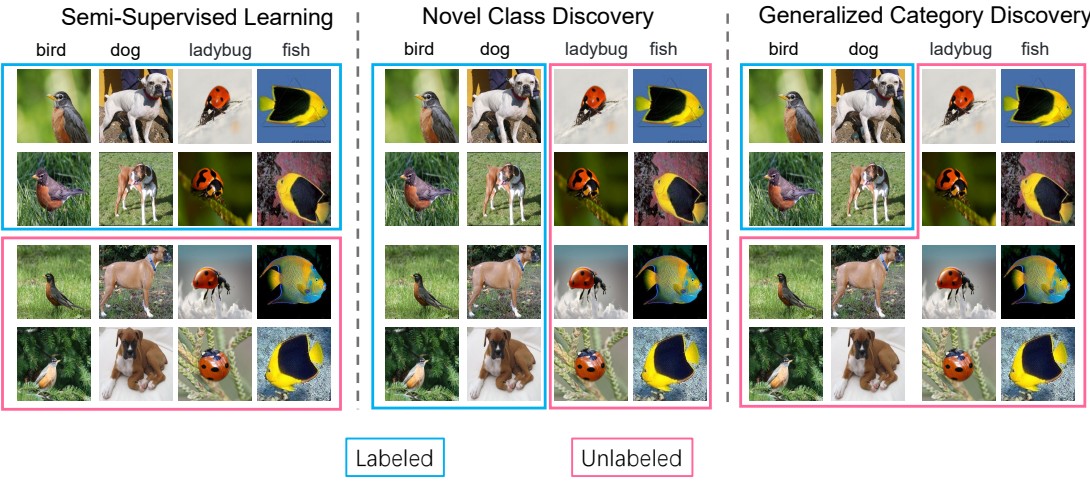

Figure 8. Difference in dataset split among SSL, NCD, and GCD.

## C. Loss Analysis

Fig. 9 shows SimGCD retains a high supervised cross-entropy (SupCE) loss during training, which indicates the noise labels in SimGCD. In contrast, our model achieves a near-zero SupCE loss. Since we introduce an auxiliary branch, it can provide more reliable soft labels to the main branch. This effectively eliminates noisy information and enhances discrimination capabilities.

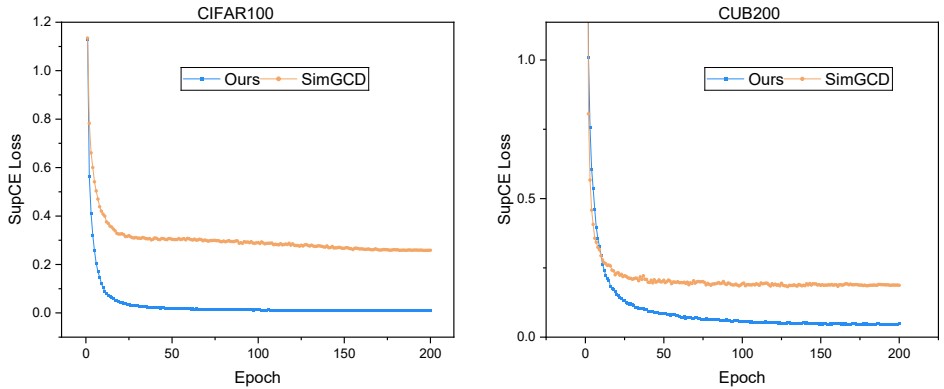

Figure 9. Supervised cross-entropy descent loss curves on CIFAR100 and CUB200.

## D. Quantitative Parameter Analysis

We provide an overview of the parameter quantities in parametric models in Table 8. Despite incorporating an additional base classifier in the auxiliary branch, our method excludes the projector, resulting in significant parameter savings. The token's contribution to the overall model size is minimal, enabling us to utilize the fewest parameters during training. During the evaluation, we abandon the base classifier and retain the extra token with the backbone, which has a negligible parameter overhead.

*Table 8.* Parameter quantity statistics among parametric models.

| Methods | Backbone | | Classifier | | Projector | | Extra | | Total |
| --- | --- | --- | --- | --- | --- | --- | --- | --- | --- |
| | Name | #Param. | Name | #Param. | Name | #Param. | Name | #Param. | |
| SimGCD | ViT-B/16 | 85,798,656 | All | 153,600 | MLP | 6,295,808 | None | 0 | 92,248,064 |
| LegoGCD | ViT-B/16 | 85,798,656 | All | 153,600 | MLP | 6,295,808 | None | 0 | 92,248,064 |
| SPTNet | ViT-B/16 | 85,798,656 | All | 153,600 | MLP | 6,295,808 | Prompts | 105,120 | 92,353,184 |
| **RLCD (Ours)** | ViT-B/16 | 85,798,656 | All+Base | 230,400 | None | 0 | Token | 768 | 86,029,824 |

## E. Comparison with Different Uncertainty Weights.

As depicted in Equation (5), we adopt the maximum probability in the auxiliary branch $\max\left(\boldsymbol{p}_b^{\mathrm{aux}}\right)$ to denote the uncertainty weight of the pseudo-base samples. Here we make a comparison with different uncertainty weights. $\boldsymbol{c}_{\max}$ is the prototype associated with the maximum probability. When the uncertainty weight is set to 0, distillation is excluded, resulting in reduced base accuracy. Conversely, the weight of 1 biases the model towards base classes, impairing novel class performance. Using the maximum cosine similarity for uncertainty yields similar results to using the maximum probability. Additionally, the uncertainty weight in the auxiliary branch obtains better performance, suggesting its greater reliability compared to the main branch.

*Table 9.* Comparison of different uncertainty weights.

| Uncertainty weight | CIFAR100 | | | CUB200 | | |
| --- | --- | --- | --- | --- | --- | --- |
| | All | Base | Novel | All | Base | Novel |
| 0 | 82.3 | 83.1 | 80.7 | 67.6 | 71.7 | **65.6** |
| 1 | 81.6 | **85.0** | 75.0 | 66.5 | 78.7 | 60.4 |
| $\cos\left(f\left(\boldsymbol{x}_i\right), \boldsymbol{c}_{\max}\right)$ | 82.0 | 83.6 | 78.9 | 68.6 | 76.9 | 64.5 |
| $\cos\left(f^{\mathrm{aux}}\left(\boldsymbol{x}_i\right), \boldsymbol{c}_{\max}^{\mathrm{aux}}\right)$ | 82.8 | 83.4 | 81.6 | 69.4 | 78.7 | 64.8 |
| $\max\left(\boldsymbol{p}_b\right)$ | 82.6 | 83.8 | 79.9 | 68.3 | 76.6 | 64.2 |
| $\max\left(\boldsymbol{p}_b^{\mathrm{aux}}\right)$ | **83.4** | 84.2 | **81.9** | **70.0** | **79.1** | 65.4 |

## F. Discussion with CRNCD

As CRNCD (Gu et al., 2023) and our approach both involve distillation, there are several key differences, listed below.

- **Different tasks.** CRNCD aims to deal with novel class discovery (NCD), where all unlabeled data belong to novel classes. In contrast, our focus is on GCD, where unlabeled data comprises both novel and base classes. Due to the intrinsic difference between these two tasks, CRNCD demonstrates unsatisfactory performance in GCD.

- **Different motivations for using distillation.** The distillation in CRNCD aims to improve novel class performance, whereas our distillation is intended to promote base class discrimination.

- **Different training paradigms.** CRNCD adopts a two-stage training procedure that first trains a supervised model and then freezes it in the second stage. Contrarily, our framework adopts one-stage training in which the main and auxiliary branches help each other simultaneously.

- **Different distilled data.** While CRNCD distills all unlabeled data, our approach focuses on pseudo-base data. Here, pseudo-base refers to predictions belonging to the base classes within the main branch.

- **Different distillation weights.** CRNCD adopts a learnable weight function to control the distillation strength. We utilize the maximum auxiliary probability as an uncertainty-based weight, which provides a simpler yet effective mechanism for regulating distillation.

Besides the above statement, we conduct a thorough experiment to compare the different distillation strategies. Table 10 shows that distilling across all unlabeled data reduces novel-class performance, as novel data increases the likelihood of incorrect base-class prediction. Additionally, the learnable distillation weight in CRNCD performs poorly for GCD, causing a significant drop in performance. These results highlight the effectiveness of our proposed design.

*Table 10.* Comparison of different distillation strategies.

| Distillation strategy | CIFAR100 | | | CUB200 | | |
|---|---|---|---|---|---|---|
| | All | Base | Novel | All | Base | Novel |
| Distill on all unlabeled data | 81.8 | 83.8 | 77.8 | 66.4 | 75.4 | 61.8 |
| Learnable weight | 80.9 | 82.6 | 77.5 | 65.0 | 74.5 | 60.2 |
| **RLCD (Ours)** | **83.4** | **84.2** | **81.9** | **70.0** | **79.1** | **65.4** |

## G. Extended Experiment on Estimated Category Numbers.

Since the estimated category numbers from GCP (Zhao et al., 2023a) cover the partial datasets, we conduct extra comparisons on other datasets under CMS (Choi et al., 2024) estimation. Table 11 shows that our method delivers better performance than CMS.

*Table 11.* Comparison of estimated category numbers on CIFAR100, FGVC-Aircraft, Herbarium-19 and Stanford Cars datasets.

| Methods | $K$ | CIFAR100 | | | FGVC-Aircraft | | | Herbarium-19 | | | Standford Cars | | |
|---|---|---|---|---|---|---|---|---|---|---|---|---|---|
| | | All | Base | Novel | All | Base | Novel | All | Base | Novel | All | Base | Novel |
| CMS (Choi et al., 2024) | 97/98/666/152 | 79.6 | 83.2 | **72.3** | 55.2 | 60.6 | 52.4 | 37.4 | 56.5 | 27.1 | 51.7 | 68.9 | 43.4 |
| **RLCD (Ours)** | 97/98/666/152 | **80.0** | **84.3** | 71.4 | **69.4** | **78.5** | **64.8** | **46.0** | **62.3** | **37.2** | **56.6** | **73.7** | **48.4** |

## H. Qualitative Visualization

As shown in Fig. 10, we utilize $t$-SNE (Van der Maaten & Hinton, 2008) to visualize the feature distribution between DINO, SimGCD, LegoGCD, and our model on the FGVC-Aircraft dataset. The $t$-SNE results for DINO demonstrate poor clustering performance, primarily due to the significant domain gap between the ImageNet and Aircraft datasets, which hinders effective feature learning. Meanwhile, SimGCD and LegoGCD achieve unsatisfactory feature clustering. It is observed that "Class 3" features of SimGCD spread out in the feature space, while LegoGCD forms two clusters of "Class 3". In contrast, our model reveals distinct clusters corresponding to different categories. The visualization comparison validates the superior feature representation of our model.

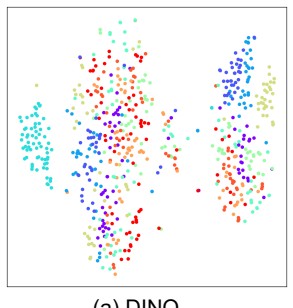 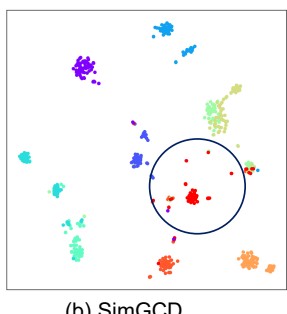 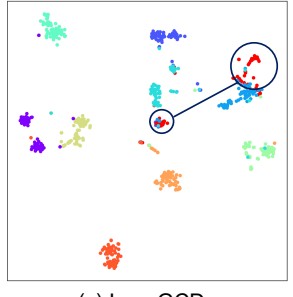 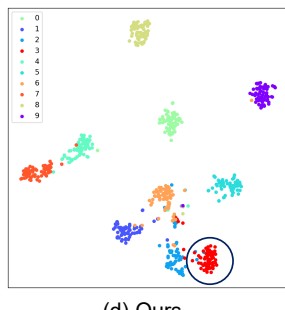

| (a) DINO | (b) SimGCD | (c) LegoGCD | (d) Ours |

*Figure 10.* $t$-SNE visualization comparing DINO, SimGCD, LegoGCD, and our method on the FGVC-Aircraft dataset, with samples randomly selected from 10 classes.

# I. Further Analysis on $H$

By definition, $H$ encourages more diverse predictions in the mini-batch. In fact, $H$ plays an important role in balancing base and novel class performance in the parametric-based method. However, $H$ would hurt base class discrimination, resulting in degraded oracle base class accuracy. Here, we conduct a deep analysis of the effect of $H$ utilizing class-wise prediction distribution (CPD). Root Mean Squared Error (RMSE) is to quantify the class prediction distribution deviation from the Ground Truth.

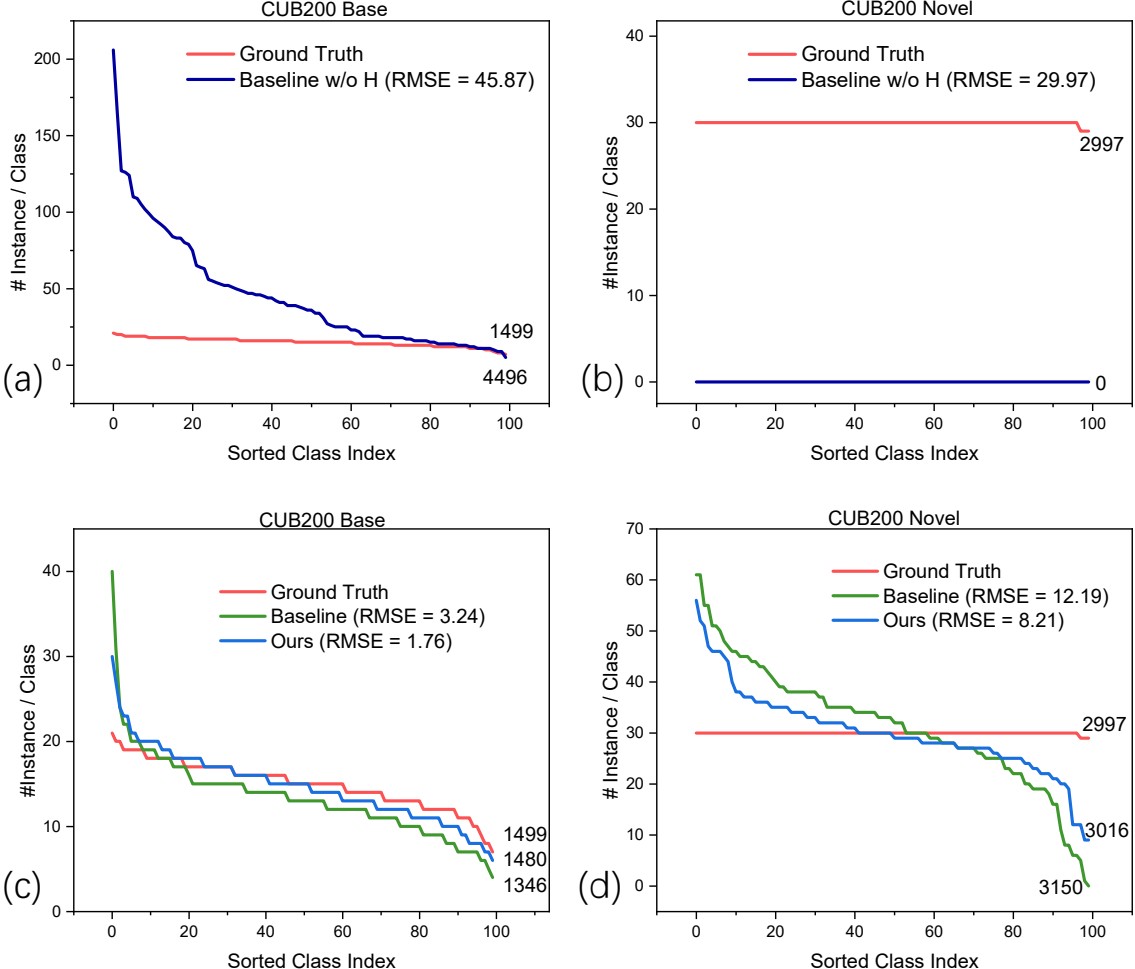

*Figure 11.* Class-wise prediction distributions on different methods. Concretely, Root Mean Squared Error (RMSE) is to measure the prediction distribution deviation from Ground Truth, and the cumulative number is marked at the end of each curve.

As shown in Fig. 11, we compare several methods, including a parametric-based baseline with and without $H$, as well as our proposed RLCD. When $H$ is removed, all samples are predicted as the base class, resulting in a significantly large RMSE for both base and novel CPDs, 45.87 and 29.97, respectively. From Fig. 11 (c) and (d), we observe that $H$ effectively refines the CPD, reducing the RMSE by 42.63 for base classes and 17.78 for novel classes. However, $H$ also introduces the side effect of misclassifying some base class samples. Specifically, the predicted base class samples are much lower than the ground truth, dropping from 1499 to 1346, which introduces noisy label learning during training. To mitigate this noisy learning, we propose a reciprocal learning framework, where the auxiliary branch provides more reliable pseudo labels to the main branch. Through cross-branch distillation, our method increases the number of predicted base class samples from 1346 to 1480. Furthermore, our CPD is closer to the ground truth, outperforming the baseline with RMSE reductions of 1.48 and 3.98 for the base and novel classes, respectively.

## J. Future Work

**Domain generalized category discovery.** Extending GCD to handle domain shifts (Wang et al., 2025) between training and testing data remains an open challenge. Future work could explore domain adaptation techniques to improve model generalization across different domains while maintaining the ability to discover novel categories.

**Continual generalized category discovery.** The current GCD setting assumes a static set of novel classes. A promising direction is to develop continual learning approaches (Zhao et al., 2023b; 2024) that can incrementally discover and learn new categories over time, while preserving knowledge of previously seen classes.

**Generalized category discovery with limited supervision.** While current GCD methods require a substantial amount of labeled base class data, real-world scenarios often have very limited labeled data. Future work could investigate transductive few-shot learning approaches to reduce the dependency on labeled data while maintaining effective novel class discovery.

