# OpenReview forum: "Generalized Category Discovery via Reciprocal Learning and Class-Wise Distribution Regularization"
_ICML.cc/2025/Conference — ICML 2025 poster_

### Official Review · Reviewer_UDum · 2025-02-22

**Overall Recommendation:** 4

**Summary:**

This paper aims to solve Generalized Category Discovery (GCD). This task seeks to discover both known and novel categories from unlabeled data, leveraging another labeled dataset with only known categories.  Different from previous works that mainly aim to boost model performance on novel categories, this work mainly aims to improve model performance on known categories. To do so, the authors introduce an auxiliary branch with a trainable token to distill known categories for the main branch. Furthermore, the authors introduce Class-wise distribution Regularization (CDR) to mitigate the leaning bias toward known categories. Experiments on wildly-used benchmarks validate the effectiveness of the proposed method.

**Claims And Evidence:**

1. The claim that previous methods struggle with identifying known categories is validated through experiments in Fig. 3.
2. The proposed method aims to boost model performance on known categories, however, the performance gain for known categories is not obvious on some benchmarks (e.g., cifar-10 and cifar-100). The authors should explain the reason for this phenomenon.

**Essential References Not Discussed:**

Missing reference:

[1] Flipped classroom: Aligning teacher attention with student in generalized category discovery. NIPS 2024
[2] Happy: A Debiased Learning Framework for Continual Generalized Category Discovery. NIPS 2024

**Experimental Designs Or Analyses:**

The experimental designs and analyses can validate the effectiveness of the proposed method.

**Methods And Evaluation Criteria:**

1. For the method, adding an extra branch to provide additional self-supervised signals for known categories is intuitive. However, the authors should explain how to guarantee the quality of pseudo labels from the additional branch. So experiments about the accuracy of the pseudo labels from the additional branch should be conducted.
2. For the evaluation, the used benchmarks and evaluation criteria are representative.
3. What's the difference between the proposed 'base class accuracy' and the 'base' in the result tables?

**Other Comments Or Suggestions:**

1. I suggest the authors to move Fig.3 foward. It is far away from the claim in Line 062 now.

**Other Strengths And Weaknesses:**

The proposed model is novel and the experiments validate the effectiveness of the proposed method.

**Questions For Authors:**

1. Experiments now assume half of the samples from known categories are labeled, what about model performance with fewer labeled samples (e.g., 25% of samples from known categories are labeled)? And what about model performance with more novel categories (e.g., 75% of categories are assumed to be novel)?

**Relation To Broader Scientific Literature:**

This paper investigates GCD from a different perspective by investigating model performance on known categories.

**Theoretical Claims:**

Theorem 3.1 is from a previous work and Theorem 3.2 is provided with detailed proof.

---

> ### Author Rebuttal · Authors · 2025-04-01
>
> **Q1:** Experiments about the accuracy of the pseudo labels from the additional branch should be conducted.
>
> **A1:** Thank you for this valuable suggestion. We have conducted a comprehensive analysis of pseudo-label accuracy across all datasets, comparing our auxiliary branch (AUX), main branch (CLS), and the SimGCD baseline:
>
> |  | CIFAR10 | CIFAR100 | ImageNet100 | CUB200 | SCars | Aircraft | Herbarium19 |
> | --- | --- | --- | --- | --- | --- | --- | --- |
> | SimGCD | 98.4 | 83.6 | 95.4 | 80.5 | 80.7 | 72.8 | 68.6 |
> | RLCD (CLS) | 98.4 | 86.3 | 95.6 | 86.9 | 90.2 | 75.5 | 76.3 |
> | RLCD (AUX) | **98.6** | **87.1** | **96.2** | **88.3** | **91.4** | **75.9** | **76.4** |
>
> These results demonstrate that our auxiliary branch consistently produces **more accurate pseudo-labels** across all datasets, with particularly significant improvements on fine-grained datasets (CUB200: +7.8%, SCars: +10.7%).  We will incorporate the analysis into our revision.
>
> **Q2:** What's the difference between the proposed 'base class accuracy' and the 'base' in the result tables.
>
> **A2:** Thank you for the insightful comment.
>
> The "oracle base accuracy" (OB) metric refers to pseudo-label accuracy of base class prediction. OB only considers the base class logits and excludes novel class prediction, providing a direct assessment of the model's discrimination of base classes.
>
> The "Base" column in our results tables reports the standard clustering accuracy on base classes after applying the Hungarian algorithm for label assignment.
>
>
> **Q3:** The performance gain for known categories is not obvious on some benchmarks (e.g., CIFAR-10 and CIFAR-100).
>
> **A3:** Thank you for this insightful observation. The varying performance gains across datasets can be attributed to several factors:
>
> - **Dataset Complexity of CIFAR-10**:  CIFAR-10 is a simple classification task with near-optimal existing results (>97%), leading to a ceiling effect. Its 10 coarse-grained classes also yield high prediction confidence, limiting the impact of our CDR component.
>
> - **Methodological Differences of CIFAR-100**: While our approach falls slightly behind CMS on CIFAR-100 base classes, CMS employs a fundamentally different paradigm (pretraining and clustering) without an all-class parametric classifier. This design choice naturally favors base class performance but limits novel class generalization, where our method demonstrates superior results.
>
> We would like to highlight that RLCD is designed to achieve balanced performance across both base and novel classes rather than optimizing for base classes alone. The above discussion will be included in our revision.
>
>
>
> **Q4:** What about model performance with fewer labeled samples (e.g., 25% of samples from known categories are labeled)? And what about model performance with more novel categories (e.g., 75% of categories are assumed to be novel)?
>
> **A4:** Thank you for this excellent suggestion. We have conducted extensive experiments varying both the proportion of labeled samples and the ratio of novel categories on the CUB200 dataset:
>
> **Varying Labeled Sample Proportion:**
>
> |  | CUB (50% labeled) |  |  | CUB (25% labeled) |  |  | CUB (10% labeled) |  |  |
> | --- | ---: | --- | :--- | ---: | --- | :--- | ---: | --- | :--- |
> |  | All | Base | Novel | All | Base | Novel | All | Base | Novel |
> | SimGCD | 60.3 | 65.6 | 57.7 | 51.0 | 52.8 | 49.7 | 34.6 | 31.8 | 37.3 |
> | RLCD | **70.0** | **79.1** | **65.4** | **63.8** | **67.5** | **61.1** | **46.7** | **45.3** | **48.0** |
> | Improvement | +9.7 | +13.5 | +7.7 | +12.8 | +14.7 | +11.4 | +12.1 | +13.5 | +10.7 |
>
> **Varying Novel Category Proportion:**
>
> |  | CUB (50% novel) |  |  | CUB (60% novel) |  |  | CUB (75% novel) |  |  |
> | --- | ---: | --- | :--- | ---: | --- | :--- | ---: | --- | :--- |
> |  | All | Base | Novel | All | Base | Novel | All | Base | Novel |
> | SimGCD | 60.3 | 65.6 | 57.7 | 56.2 | 65.1 | 53.3 | 51.8 | 68.3 | 49.1 |
> | RLCD | **70.0** | **79.1** | **65.4** | **62.5** | **75.6** | **58.1** | **56.8** | **78.1** | **53.2** |
> | Improvement | +9.7 | +13.5 | +7.7 | +6.3 | +10.5 | +4.8 | +5.0 | +9.8 | +4.1 |
>
> These results reveal several important insights:
>
> - **Label Efficiency**: Reducing labeled data degrades performance, yet our RLCD maintains substantial performance advantages even with severely limited labeled data (10% labeled), demonstrating its superiority in low-resource scenarios.
>
> - **Scalability to Novel Classes**: As the proportion of novel classes increases, our method still outperforms SimGCD across all metrics, though the margin narrows for novel classes. This is expected as the clustering problem becomes more challenging with more novel classes. The strong performance on base classes is sustained primarily due to the limited class number.
>
>
> **Q5:** Missing literatures: FlipClass (NeurIPS 2024), Happy (NeurIPS 2024).
>
> **A5:** Thank you for bringing these works to our attention. We will incorporate the suggested references in our revised manuscript.

---

> > ### Comment · Reviewer_UDum · 2025-04-02
> >
> > Thanks for your responses, I have increased my score.

---

> > > ### Author Response · Authors · 2025-04-02
> > >
> > > We sincerely appreciate your recognition of our work and the improved rating.

---

### Official Review · Reviewer_bo6h · 2025-03-12

**Overall Recommendation:** 3

**Summary:**

This manuscript addresses the Generalized Category Discovery (GCD) task, which aims to classify unlabeled data containing both existing (base) and novel (unknown) classes. Existing parametric methods often compromise the discriminability of known classes in order to identify novel classes. To remedy this shortcoming, the authors propose a single-stage Reciprocal Learning Framework (RLF) and a Class-wise Distribution Regularization (CDR) approach.

**Claims And Evidence:**

Yes

**Essential References Not Discussed:**

No

**Experimental Designs Or Analyses:**

Yes

**Methods And Evaluation Criteria:**

Yes

**Other Comments Or Suggestions:**

None

**Other Strengths And Weaknesses:**

**Strengths**:

​	1.	The manuscript is well-written and logically coherent.

​	2.	The proposed reciprocal learning framework is simple and effective, introducing constraints on old/known classes in a way that enhances performance.



**Weaknesses**:

​	1.	It lacks a comparison with the latest state-of-the-art method, FlipClass [1].

​	2.	Because CDR penalizes distribution-level discrepancies between different views, the model tends to produce more balanced class predictions rather than heavily favoring any single class. If one class truly occupies a very small portion of the dataset, the model may naturally lean toward majority classes across several batches. Consequently, CDR might forcibly boost predictions for minority classes, leading to a certain degree of over-correction. In other words, it does not incorporate prior knowledge of strongly skewed class distributions; it merely aims to ensure sufficient agreement and sharpness in class-level predictions from two different views. In the presence of severely long-tailed or extremely imbalanced data, this can introduce errors. The authors do not analyze this scenario in detail.

**Questions For Authors:**

Please see weaknesses.

**Relation To Broader Scientific Literature:**

No

**Theoretical Claims:**

Yes

---

> ### Author Rebuttal · Authors · 2025-04-01
>
> **Q1.** It lacks a comparison with the latest state-of-the-art method, FlipClass [1].
>
> **A1.** Thank you for highlighting this important work. We observed that FlipClass employs asymmetric augmentation and mixup techniques for performance enhancement from their supplementary materials. For a fair comparison, we incorporated similar advanced augmentation strategies into our approach, resulting in further improvements across datasets:
>
> |  | CIFAR10 | CIFAR100 | ImageNet100 | CUB200 | SCars | Aircraft | Herbarium-19 |
> | --- | --- | --- | --- | --- | --- | --- | --- |
> | FlipClass | **98.5** | 85.2 | 86.7 | 71.3 | 63.1 | 59.3 | 46.3 |
> | RLCD | 98.2 | **85.6** | **87.2** | **72.5** | **65.7** | **62.2** | **47.5** |
>
> The results demonstrate that our method outperforms FlipClass on 6 out of 7 GCD datasets, with particularly significant improvements on fine-grained datasets (SCars: +2.6%, Aircraft: +2.9%).
>
> **Q2.** Because CDR penalizes distribution-level discrepancies between different views, the model tends to produce more balanced class predictions rather than heavily favoring any single class. If one class truly occupies a very small portion of the dataset, the model may naturally lean toward majority classes across several batches. Consequently, CDR might forcibly boost predictions for minority classes, leading to a certain degree of over-correction. In other words, it does not incorporate prior knowledge of strongly skewed class distributions; it merely aims to ensure sufficient agreement and sharpness in class-level predictions from two different views. In the presence of severely long-tailed or extremely imbalanced data, this can introduce errors. The authors do not analyze this scenario in detail.
>
> **A2.** We appreciate this thoughtful analysis of CDR and are pleased to address these concerns with both theoretical insights and empirical results:
>
> - **Implicit adaptive weighting in CDR:** In long-tailed distributions, majority classes typically yield higher-confidence predictions [2], leading to lower assigned probabilities for minority classes. According to class-wise distribution definition, $m_k=\frac{1}{\sum_{i=1}^N p_i^{(k)}}\left(\sum_{i=1}^N p_i^{(k)} p_i\right)$, this results in **reduced weighting** of high-confidence majority class samples when computing minority class distributions. Consequently, CDR naturally adjusts sample contributions, establishing an adaptive weighting mechanism that helps **mitigate over-correction**.
>
> - **Balanced learning via CDR and CE**: Our RLCD combines cross-entropy (CE) loss with CDR. While standard CE training in long-tailed settings often biases predictions toward majority classes [2], CDR counteracts this by promoting balanced learning across all classes, **mitigating excessive skew**. This is quantitatively demonstrated in Fig. 11, where our method reduces the Root Mean Square Error (RMSE) by 1.48 and 3.98 for base and novel classes, respectively, compared to the baseline.
>
> -  **Strong performance in the long-tailed setting**: Our RLCD achieves state-of-the-art results on the Herbarium-19 dataset, which features a naturally long-tailed distribution. As reported in Table 1, RLCD surpasses prior methods by **1.3% overall** and **2.5% on base classes**, demonstrating robustness in such scenarios.
>
> Furthermore, we would like to emphasize that RLCD leverages both the Reciprocal Learning Framework (RLF) and CDR. As shown in Tables 1 and 2, the experimental results demonstrate its versatility across **generic**, **fine-grained**, and **long-tailed** scenarios, consistently delivering superior performance.
>
> [1] Lin H, An W, Wang J, et al. Flipped classroom: Aligning teacher attention with student in generalized category discovery. NeurIPS, 2024.\
> [2] Menon A K, Jayasumana S, Rawat A S, et al. Long-tail learning via logit adjustment. ICLR, 2021.

---

> > ### Comment · Reviewer_bo6h · 2025-04-06
> >
> > Thanks for the detailed reply, which addressed all my concerns. Thus, I have increased my score.

---

> > > ### Author Response · Authors · 2025-04-06
> > >
> > > Thank you for your kind review and improved rating. We are pleased to address your concerns.

---

### Official Review · Reviewer_2AkZ · 2025-03-12

**Overall Recommendation:** 2

**Summary:**

The paper proposes a novel approach for Generalized Category Discovery by introducing a Reciprocal Learning Framework and Class-wise Distribution Regularization. RLF enhances base class discrimination through an auxiliary branch that distills reliable base-class predictions to the main branch, while CDR mitigates learning bias toward base classes by enforcing consistency in class-wise probability distributions. Experiments on GCD benchmarks demonstrate state-of-the-art performance, with significant improvements in both base and novel class accuracy.

**Claims And Evidence:**

RLF improves base discrimination via cross-branch distillation. While ablation studies show gains, the mechanism of reliable soft labels from the auxiliary branch is not validated, and there is no analysis of pseudo-label accuracy or error rates.

**Essential References Not Discussed:**

N/A

**Experimental Designs Or Analyses:**

Critical parameters ($ \alpha, \beta$) are set to 0.5 without sensitivity analysis. Fig. 6 shows performance variation but lacks justification for the chosen values.
The impact of removing InfoNCE (Sec. 3.1) is not isolated. Table 3 conflates multiple components, making it unclear which contributes most.

**Methods And Evaluation Criteria:**

The auxiliary branch design of RLF is intuitive but under-explored and freezing the shared transformer block may limit flexibility.
Benchmarks are standard, but the ACC metric alone is insufficient. Clustering metrics (NMI, ARI) and robustness to class imbalance are not reported. The assumption of known total class count $ |Y_u| $ is impractical; results with estimated $ K $ (Table 4) show significant performance drops but are not discussed thoroughly.

**Other Comments Or Suggestions:**

Fig. 1’s caption is unclear; "base logits" and "all logits" need explicit definitions.

**Other Strengths And Weaknesses:**

**Strengths**:
Practical design with negligible inference overhead.
Improved base-class performance validated across datasets.

**Weaknesses**:
RLF is an incremental extension of multi-branch SSL, and CDR resembles distribution alignment in domain adaptation.
While performance gains are clear, the method does not address the core challenges of GCD (unknown class counts, domain shift).

**Questions For Authors:**

1. How does RLF perform *without* the shared transformer block? Does freezing the block limit novel-class adaptation?
2. Table 4 shows performance drops with estimated $K $. Can CDR be adapted to handle noisy $K $?
3. Can Theorem 3.2 be extended to non-ideal cases (when $ m_k \neq m'_k $)?

**Relation To Broader Scientific Literature:**

The work builds on parametric GCD (SimGCD, LegoGCD) and SSL but does not discuss connections to *open-world semi-supervised learning* or *multi-task distillation*. The auxiliary branch resembles multi-exit networks, but prior art is unmentioned.

**Theoretical Claims:**

Theorem 3.2’s proof (Appendix A.1) correctly uses Cauchy-Schwarz but assumes $ m_k$ and $ m'_k $ are exactly aligned, which is unrealistic in practice. The link between CDR and "boosting novel performance" (Sec. 3.3) is not theoretically justified.

---

> ### Author Rebuttal · Authors · 2025-04-01
>
> **Q1.** There is no analysis of pseudo-label accuracy.
>
> **A1.** Please refer to our response to Reviewer UDum-Q1.
>
> **Q2.** Fig. 6 shows the performance variation of parameters α and β but lacks justification for the chosen values.
>
> **A2.** Parameters α and β control the strength of distillation and regularization, respectively. Since the supervised loss weight λ=0.35 in the baseline (SimGCD), we explore values in the range [0,1] to match a comparable scale.
>
> **Q3.** The impact of removing InfoNCE (Sec. 3.1) is not isolated.
>
> **A3.** We have conducted additional experiments to isolate the impact of InfoNCE:
>
> |  | CIFAR100 |  |  | CUB200 |  |  |
> | --- | ---: | --- | :--- | ---: | --- | :--- |
> |  | All | Base | Novel | All | Base | Novel |
> | RLCD + InfoNCE | 82.8 | 83.4 | 81.6 | 68.9 | 76.7 | 65.0 |
> | RLCD | **83.4** | **84.2** | **81.9** | **70.0** | **79.1** | **65.4** |
>
> Removing InfoNCE consistently improves performance across all metrics. This is because InfoNCE tends to push apart same-class features, which impairs feature discrimination.
>
> ---
>
> **Q4.** Table 3 conflates multiple components, making it unclear which contributes most.
>
> **A4.**  Table 3 presents the contribution of each component. The important facts are that RLF benefits base class performance, while CDR significantly improves novel class performance.
>
> **Q5.** RLF is an incremental extension of multi-branch SSL, and CDR resembles distribution alignment in domain adaptation.
>
> **A5.** We would like to highlight several key distinctions:
>
>  - **Lightweight Architectural Design:** Unlike traditional multi-branch SSL methods (e.g., MoCo, BYOL) that require separate networks with significant memory and computational overhead, our approach inserts a **single auxiliary token** in the last transformer block.
>
>  - **Cross-View Consistency vs. Cross-Domain Alignment:**  CDR is inherently different from domain adaptation methods. CDR operates on **different views** of the same samples to **increase prediction confidence** by enforcing cross-view consistency. This contrasts with domain adaptation, which aims to align distributions across distinct domains.
>
> ---
> **Q6.** While performance gains are clear, the method does not address the core challenges of GCD (unknown class counts, domain shift).
>
> **A6.** Our work primarily focuses on improving standard GCD performance, following recent approaches like SimGCD (ICCV 2023) and LegoGCD (CVPR 2024). Regarding the challenges:
>
> -  **Unknown Class Counts**: From our response to Reviewer 54Bx-Q2, our method can refine the estimation through a multi-stage process.
>
> -  **Domain Shift**: While domain shift is a challenge in GCD, methods like HiLo (ICLR2025) address it by leveraging domain adaptation techniques.  We plan to extend our approach to domain-shift scenarios in future work.
>
>
>
> **Q7.** Fig. 1's caption is unclear; "base logits" and "all logits" need explicit definitions.
>
> **A7.** We will clarify the caption in our revision. "Base logits" refers to the output probabilities for base classes only, while "all logits" include the output probabilities for both base and novel classes.
>
> ---
> **Q8.** How does RLF perform without the shared transformer block? Does freezing the block limit novel-class adaptation?
>
> **A8.** We conducted additional experiments comparing three configurations:
>
> |  | CIFAR100 |  |  | CUB200 |  |  |
> | --- | ---: | --- | :--- | ---: | --- | :--- |
> |  | All | Base | Novel | All | Base | Novel |
> | Shared block | **83.4** | **84.2** | 81.9 | **70.0** | **79.1** | 65.4 |
> | Separate block | 83.1 | 83.7 | **82.2** | 69.5 | 76.6 | **66.0** |
> | Frozen block | 80.5 | 80.7 | 80.2 | 63.3 | 71.3 | 59.4 |
>
> Using a separate block leads to a **slight performance drop**, as sharing the transformer block allows for learning more robust, task-generalizable parameters. Freezing the block, however, causes a **substantial decline** across all metrics, underscoring the importance of fine-tuning for effective adaptation to both base and novel classes. This aligns with **common practice** in GCD works, where fine-tuning the final block is critical to address domain shifts in downstream datasets.
>
> **Q9.** Table 4 shows performance drops with estimated K. Can CDR be adapted to handle noisy K?
>
> **A9.** Our method inherently handles noisy class number estimates through its multi-stage refinement process, as detailed in our response to Reviewer 54Bx-Q2.
>
> **Q10.** Can Theorem 3.2 be extended to non-ideal cases (when mk≠mk′)?
>
> **A10.**  We would clarify $m_k ≠ m_{k'}$ is **indeed the common scenario**, as these distributions come from different views of batch samples. Theorem 3.2 characterizes the **optimization objective** rather than the **initial condition**.  It aims to maximize distribution similarity between the two views while approximating one-hot.
>
> ---
> We sincerely appreciate the reviewer’s effort in evaluating our paper and hope that our responses adequately address the concerns.

---

### Official Review · Reviewer_54Bx · 2025-03-12

**Overall Recommendation:** 4

**Summary:**

This paper studies the task of Generalized Category Discovery (GCD). It builds upon parametric-based GCD methods, and proposes a Reciprocal Learning Framework (RLF) that introduces an auxiliary branch devoted to base classification. Within the framework, the main branch filters the pseudo-base samples to the auxiliary branch while the auxiliary branch provides more reliable soft labels for the main branch, leading to a virtuous cycle. The paper further incorporates Class-wise Distribution Regularization (CDR) to mitigate the leaning bias towards base classes. Experiments validate the superiority of the proposed method.

**Claims And Evidence:**

Yes.

**Essential References Not Discussed:**

Please include more recent papers [R1,R2,R3] in GCD to reflect the potential trends of this field.

References:
[R1]. Active Generalized Category Discovery. CVPR 2024.
[R2]. Federated Generalized Category Discovery. CVPR 2024.
[R3]. Happy: A Debiased Learning Framework for Continual Generalized Category Discovery.

**Experimental Designs Or Analyses:**

Yes.

**Methods And Evaluation Criteria:**

Yes.

**Other Comments Or Suggestions:**

Please consider making the dicussion of related work more comprehensive, as in Weakness 3.

**Other Strengths And Weaknesses:**

Strength:
1. This paper is well-motivated and easy to follow.
2. The proposed auxiliary learning with additional learnable tokens ensures the learning of the basic classes, which fundamentally improves the overall performance.
3. The performance gain is remarkable compared with prior arts.


Weaknesses:
1. Could the authors explain how the auxiliary helps the main branch intuitively, as well as the effect of Class-wise Distribution Regularization?
2. Could the method estimate the number of new classes instead of borrowing off-the-shelf results?
3. Please include more recent papers [R1,R2,R3] in GCD to reflect the potential trends of this field.

References:
[R1]. Active Generalized Category Discovery. CVPR 2024.
[R2]. Federated Generalized Category Discovery. CVPR 2024.
[R3]. Happy: A Debiased Learning Framework for Continual Generalized Category Discovery.

**Questions For Authors:**

No.

**Relation To Broader Scientific Literature:**

This paper proposes to employ both main and auxiliary branch, which ensures the basic class learning. This idea is novel to the community of GCD.

**Theoretical Claims:**

There is no theoretical analysis in this paper.

---

> ### Author Rebuttal · Authors · 2025-04-01
>
> **Q1.** Could the authors explain how the auxiliary helps the main branch intuitively, as well as the effect of Class-wise Distribution Regularization?
>
> **A1.** Thanks for your valuable feedback.
> The auxiliary branch supports the main branch in two key ways:
>
> -  **Improve Base Performance:** The auxiliary branch provides more reliable pseudo labels to guide the main branch in maintaining high base class accuracy.
>
> -  **Enhance Feature Learning:** Since both branches share the final feature extraction block but are assigned complementary tasks (base-only vs. all-class classification). Consequently, the model is encouraged to learn more robust parameters and develop better feature representations.
>
> Class-wise Distribution Regularization (CDR) serves two key purposes:
>
> - **Mitigate Prediction Bias:** Through class-wise regularization, CDR treats each class equally. Consequently, CDR reduces prediction bias toward base classes, enabling more novel class samples to be correctly classified as novel.
>
> -  **Boost Novel Performance:** CDR implicitly enhances prediction confidence, allowing the novel classifier to better match novel samples and significantly improving novel class accuracy.
>
>
>
> ---
> **Q2.** Could the method estimate the number of new classes instead of borrowing off-the-shelf results?
>
> **A2.** Thank you for this constructive comment.  The category number can be estimated using machine learning algorithms such as semi-supervised k-means and agglomerative clustering. Beyond directly adopting off-the-shelf estimates, we introduce a multi-stage refinement process that leverages our improved feature representations to **enhance estimation** precision:
>
> - **Initial Estimation:** We begin with an off-the-shelf method, such as semi-supervised k-means, to establish a baseline estimate.
> - **Refinement:** After training, we re-estimate the number of classes using the enhanced feature representations from our model.
> - **Retraining:** The refined estimation is then used to retrain the model for improved performance.
>
> We validate this approach on ImageNet-100, CUB200, and SCars, with results shown below:
>
> |  | ImageNet-100 (K: 100) |  |  | CUB200 (K: 200) |  |  | SCars (K: 196) |  |  |
> | --- | ---: | --- | :--- | ---: | --- | :--- | ---: | --- | :--- |
> |  | All | Base | Novel | All | Base | Novel | All | Base | Novel |
> | Est. K (1st) 109/231/230 | 84.4 | 93.2 | 80.0 | 68.4 | 77.1 | 64.1 | 58.6 | 76.4 | 50.8 |
> | Est. K (2nd) 106/218/212 | **85.3** | **93.1** | **81.4** | **69.2** | **76.3** | **65.7** | **59.9** | **77.6** | **51.3** |
>
> The table shows that the second estimation **more closely** approximate the ground-truth K, consistently enhancing performance across datasets.
> The improved estimation stems from our model’s **superior feature representation**, which also manifests in clearer class separation in the t-SNE visualizations (see Appendix A.8). Notably, clustering-based methods like GCD and CMS cannot leverage this multi-stage refinement, as their training excludes parametric classifiers.
>
> **Q3.** Please include more recent papers [R1, R2, R3] in GCD to reflect the potential trends of this field.
>
> **A3.** We appreciate this suggestion and will incorporate these recent papers in our revision to provide a more comprehensive view of the field's development.

---

### Decision · Program_Chairs · 2025-05-01

**Decision:**

Accept (poster)

**Comment:**

This paper initially received mixed reviews: 2 * Weak Reject and 2 * Weak Accept. The authors submitted a rebuttal addressing the reviewers’ concerns and questions. After considering the rebuttal and subsequent discussion, three reviewers were satisfied with the authors’ responses and raised their scores to 1 * Weak Accept and 2 * Accept. Although Reviewer 2AkZ acknowledged that some concerns were addressed, they felt the paper still needed improvement in terms of the motivation and detailed presentation, and thus maintained their original score of Weak Reject.

Upon reviewing the comments from Reviewer 2AkZ and the authors’ rebuttal, the AC found that: 1) Reviewer 2AkZ did not clearly point out their concerns regarding motivation and presentation details in the initial review; 2) and the authors have adequately addressed the questions and concerns that were raised.

The AC also acknowledged that Reviewer 2AkZ continued to express concern after the rebuttal that the proposed method does not directly address certain core challenges of Generalized Category Discovery (GCD), such as unknown class counts and domain shift. However, most existing GCD methods primarily focus on improving the clustering accuracy of unlabeled data, which remains a central objective of the field. Therefore, not explicitly tackling class count estimation or domain shift should not be considered a major limitation, as these aspects represent distinct research directions within the broader GCD framework.

Considering these points, the AC believes that the authors have sufficiently addressed the reviewers' concerns and that the strengths of the work outweigh the remaining minor limitation (raised by Reviewer 2AkZ). Thus, the AC recommends acceptance if room permits in the program. The AC also strongly encourages the authors to incorporate all the clarifications from the rebuttal and to further improve the clarity and presentation in the final version.